# Gestational weight gain among pregnant women in Ibadan, Nigeria: Pattern, predictors and pregnancy outcomes

Ikeola A. Adeoye[1,2]*, Elijah A. Bamgboye[1], Akinyinka O. Omigbodun[3]

1 Faculty of Public Health, Department of Epidemiology and Medical Statistics, College of Medicine, University of Ibadan, Ibadan, Nigeria, 2 Consortium of Advanced Research for Africa (CARTA), Nairobi, Kenya, 3 Faculty of Clinical Sciences, Department Obstetrics and Gynaecology, College of Medicine, University of Ibadan, Ibadan, Nigeria

* adeoyeikeola@yahoo.com, iadeoye@cartafrica.org

**Data Availability Statement:** The datasets generated and/or analysed during the current study are not publicly available because they contain potentially identifying and confidential information but are available from the UI/UCH Ethics

## Abstract

### Background

Gestational weight gain (GWG) is a risk factor for adverse pregnancy outcomes, future obesity and chronic diseases among women. However, has not received much attention in many low and middle-income countries such as Nigeria. We investigated the pattern, associated factors and pregnancy outcomes of GWG in Ibadan, Nigeria, using the Ibadan Pregnancy Cohort Study (IbPCS).

### Methodology

The IbPCS is a multicentre prospective cohort study conducted among 1745 pregnant women recruited from four health facilities in Ibadan, Nigeria. GWG, the primary outcome, was categorised according to the Institute of Medicine's classification into insufficient, adequate and excessive weight gain. Pregnancy outcomes were the secondary outcome variables. Logistic regression analysis (Adjusted odds ratios and 95% confidence interval CI) was used to examine associations, and Poisson regression analyses were used to investigate associations with outcomes.

### Results

Only 16.9% of women had optimal GWG, 56.8% had excessive GWG, and 26.9% had insufficient GWG. Excessive GWG was associated with high income '> #20,000-' (AOR: 1.64, 95% CI: 1.25–2.17), being overweight (AOR: 2.12, 95% CI: 1.52–2.95) and obese (AOR: 1.47, 95% CI: 1.02–2.13) after adjusting for confounders. In contrast, increased odds of insufficient GWG have associated women with depression (AOR: 1.70, 95% CI 1.17–2.47). There was no significant association between inappropriate GWG and pregnancy outcomes However, there was an increased odds for postpartum haemorrhage (AOR: 2.44, 95% CI 1.14–5.22) among women with obesity and excessive GWG.

Committee (uiuchec@gmail.com) on reasonable request if it meets the criteria for accessing confidential data.

**Funding:** This research was supported by the Consortium for Advanced Research Training in Africa (CARTA). CARTA is jointly led by the African Population and Health Research Center and the University of the Witwatersrand and funded by the Carnegie Corporation of New York (Grant No. G-19-57145), Sida (Grant No:54100113), Uppsala Monitoring Center, Norwegian Agency for Development Cooperation (Norad), and by the Wellcome Trust [reference no. 107768/Z/15/Z] and the UK Foreign, Commonwealth & Development Office, with support from the Developing Excellence in Leadership, Training and Science in Africa (DELTAS Africa) programme. Ikeola Adeoye is a CARTA PhD fellow. The statements made and views expressed are solely the responsibility of the Fellow. The funders had no role in study design, data collection and analysis, decision to publish, or preparation of the manuscript. For the purpose of open access, the author has applied a CC BY public copyright licence to any Author Accepted Manuscript version arising from this submission. There was no additional external funding received for this study.

**Competing interests:** The authors have declared that no competing interests exist.

## Conclusions

Excessive GWG was the most typical form of GWG among our study participants and was associated with high maternal income, and being overweight or obese. GWG needs to be monitored during antenatal care, and interventions that promote appropriate GWG should be implemented among pregnant women in Nigeria.

## Introduction

Gestational weight gain (GWG) is the weight gained during pregnancy, a physiologic response necessary for foetal growth and development and the increased metabolic demand of pregnancy [1]. Inappropriate GWG is a public health concern associated with adverse pregnancy outcomes, future obesity and chronic diseases among women. The 2009 Institute of Medicine (IOM) guideline provided specific recommendations for GWG according to BMI based on the WHO classification [2]. Precisely, underweight women, normal weight, overweight and obese are to achieve a recommended total weight gain as follows (12.5–18.0) kg, (11.5–16.0) kg, (7.0–11.0) kg, (5.0–8.0) kg respectively. Also, the recommended weight gain per week in the second and third trimesters in kg/week are underweight (0.44–0.58), normal weight (0.35–0.50), overweight (0.23–0.33) and obese (0.17–0.27) [3,4]. Therefore women who gain weight below the recommended threshold have insufficient GWG, which is associated with intrauterine growth retardation, low birth weight, prematurity, and an increased risk of neonatal morbidity and mortality [5,6]. Women who gained weight within the stipulated range are described as having adequate GWG, while those that exceed the recommendation have excessive GWG. Excessive GWG adversely affects maternal metabolism and foetal development and is associated with preeclampsia, gestational diabetes mellitus, caesarean section and macrosomia [7–10]. Its long-term complications include postpartum weight retention and perpetuating the cycle of obesity among women of reproductive age [7,11,12].

Appropriate weight gain is crucial for optimal pregnancy outcomes, and weight gained outside the recommended range and the associated adverse health outcomes will increase the health care cost, the length of hospital stay, maternal morbidity and mortality [13]. In the past, insufficient GWG had been the primary concern among pregnant women in low and middle-income countries (LMIC) [3,7]. However, the current global obesity epidemic among women of reproductive age, which has resulted from urbanisation, globalisation, and changes in food consumption patterns and physical inactivity, has also led to excessive GWG from excess caloric intake [14]. Various inappropriate GWGs have been reported in different countries, mainly from North America and Europe, some from Asia. At the same time, evidence is sparse in Africa, especially those that used the IOM classification. For example, a systematic review and meta-analysis of over one million pregnant women from 23 studies from North America and Europe reported that 23% and 47% had insufficient and excessive GWG, respectively. Other countries have also documented the prevalence of insufficient and excessive GWG, namely China (22.6% versus 50%) [15]; Brazil (22.6% versus 50%) [16]; Turkey (10.8% versus 14.0%) [17]; Cameroun (40% versus 32%) [18]. Notably, a systematic review that assessed GWG in Africa from 26 studies noted that insufficient GWG was predominant in most African countries, particularly the poorest countries [19]. In contrast, South Africa had the highest proportion of excessive GWG (55%) in the region. The predictors of GWG in Africa include maternal education, income, obtaining a minimum of four ANC visits, being physically active, revenue, and dietary pattern [19,20].

Nigeria bears a considerable burden of maternal mortality with a maternal mortality ratio of 578 maternal deaths per life birth and contributes 12% to global figures annually (35 000 maternal deaths [21]. At the same time, Nigeria is undergoing nutrition and epidemiological transitions, leading to an upsurge of obesity in the general population, including women of reproductive age [22]. The coexistence of obesity and excessive GWG leads to more severe pregnancy complications. For instance, studies among Chinese women showed that obese women with excessive GWG had much higher odds for hypertensive disorders of pregnancy than normal-weight women with adequate weight gain [23–25]. Despite this, GWG has been a neglected maternal health issue in Nigeria. Although assessing maternal weight at every antenatal visit is routine, GWG is neither evaluated nor addressed in most maternal healthcare settings in Nigeria [26,27]. This contrasts with the practice in the United States, where The American College of Obstetricians and Gynaecologists (ACOG) recommends that *"Maternal care workers, which include obstetricians, doctors, midwives and nurses should assess women's BMI at the antenatal booking visit. They also provide appropriate information, education and communication on maternal nutrition and exercise. Optimal weight and the importance of preventing excessive weight gain* [28]. Furthermore, the current body of evidence on GWG in Nigeria is limited, with only a few emerging studies available [29–32]. Asefa *et al.* (2020) noted a lack of research, the poor quality and methodology associated with GWG studies in Africa, hence suggesting the need for methodologically sound studies to address GWG-related research questions, including the related factors and pregnancy outcomes in sub-Saharan Africa [19]. Therefore we investigated the pattern, prevalence, predictors and pregnancy outcomes of GWG in Ibadan, Nigeria, using the Ibadan pregnancy cohort study.

## Materials and methods

### Study design, setting and population

This study was a component of the Ibadan Pregnancy Cohort Study (IbPCS), which aimed to investigate the associations between maternal obesity, lifestyle characteristics, glycaemic control, GWG and the pregnancy outcomes in Ibadan. The details of the methodology have been published elsewhere [33]. In summary, IbPCS is a prospective cohort study conducted among 1745 pregnant women in Ibadan, Nigeria, who were recruited early in pregnancy (GA $\leq$ 20 weeks) at their first antenatal visit at four selected health facilities in Ibadan. These facilities were University College Hospital, Adeoyo Maternity Teaching Hospital, Jericho Specialist Hospital, and Saint Mary Catholic Hospital, Oluyoro Ibadan. They provide comprehensive obstetric services to pregnant women and are the major referral centres for obstetric emergencies within the Ibadan metropolis. During recruitment, participants were assessed by trained research nurses and data were obtained on baseline information and lifestyle characteristics through personal interviews and a desktop review of medical records. The baseline information included sociodemographic characteristics, obstetric and past medical history, and specific lifestyle characteristics.

The lifestyle factors examined included dietary patterns, including sugar-sweetened beverages (SSB), physical activity, sedentary behaviour, tobacco use, alcohol consumption, and sleep pattern. Anthropometric measures like weight, height and mid-upper arm circumference were obtained. Serial weight values were used to estimate GWG according to the Institute of Medicine guidelines (2009) [4] which was the difference between the booking weight and the last weight assessed during the third trimester at the ANC (32.4 ± 4.7 weeks). After recruitment ($\leq$ 20 weeks), study participants were followed up at three time points: 24–28 weeks, third trimester, at delivery. The flowchart of study participants from enrolment to delivery is shown in Fig 1 and 1254 out of the 1745 participants were included in the final analysis.

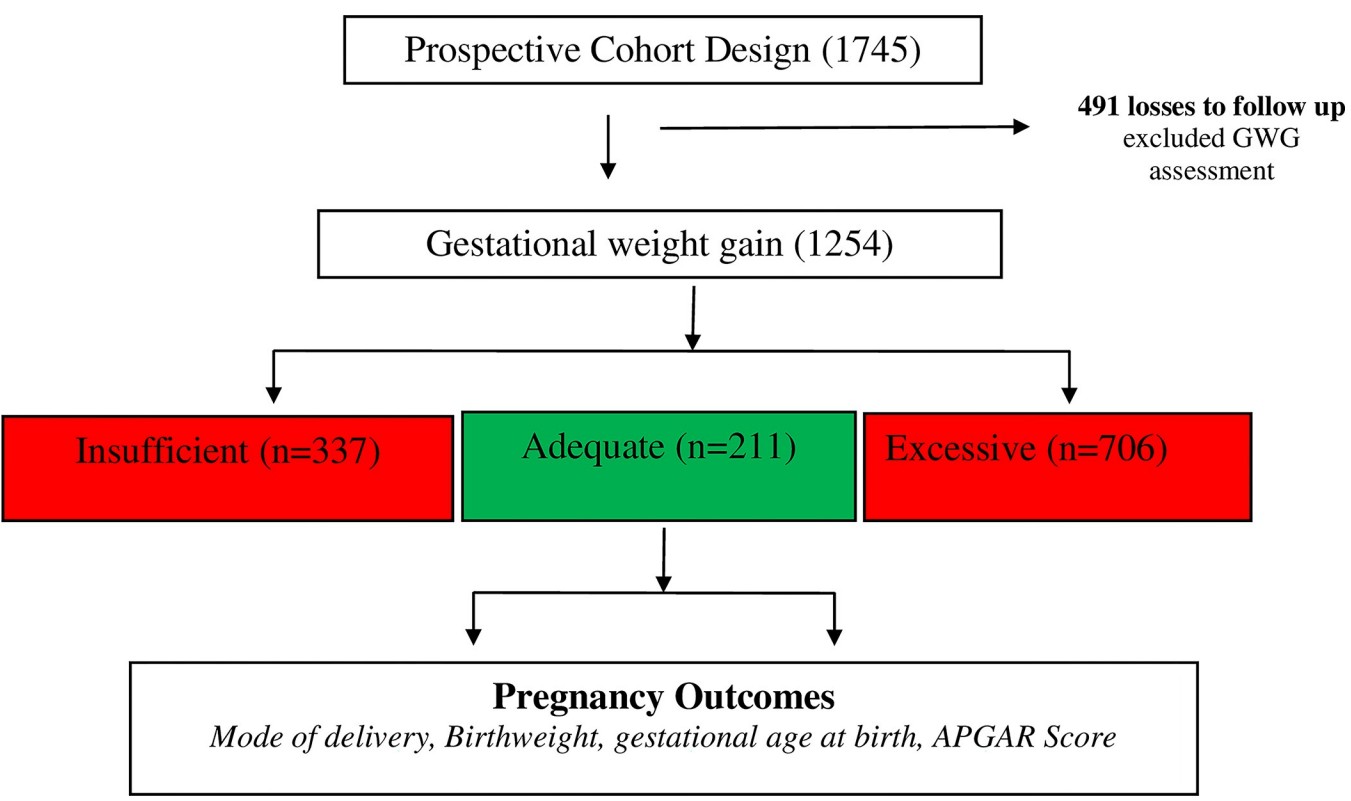

**Fig 1. Flowchart of study participants.**

**Ethical approval.** The ethical approval for this study was obtained from the University of Ibadan/University College Hospital (UI/UCH) Institutional Review Board (UI/EC/15/0060) and the Oyo State Ministry of Health Ethical Committee (AD/13/479/710). We obtained verbal and written informed consent from respondents before recruitment into the study. The study protocol and conduct adhered to the principles in the Declaration of Helsinki. Minors (participants under the age of 18 years) were not included in this study.

## Data collection procedures

**Measures.** Data were collected by trained research staff using pretested, interviewer-administered questionnaires and desktop review of medical records at booking, third trimester, and delivery. Ideally, GWG measures the difference between the weight at delivery and the prepregnancy weight. However, these were unavailable; hence the difference between the booking weight and the last third-trimester weight measured was used to estimate the GWG. The GWG rate was calculated as the difference between the last third-trimester weight recorded and the booking weight divided by the number of gestational weeks between in the interval these weights were measured as annotated as follows.

$$\text{GWG rate} = \frac{\textit{third trimester weight (final)} - \textit{weight at booking}}{\textit{GA at third trimester weight (final)} - \textit{GA at booking}}$$

The Institute of Medicine (IOM) guidelines provide recommendations for weight gain during pregnancy based on a woman's pre-pregnancy body mass index (BMI). Hence, we categorised the GWG rate into insufficient, adequate, and excessive weight gain. Insufficient GWG was

defined as below the lower BMI recommended weight gain per week, and excessive GWG as above the upper BMI recommended weight gain per week [1]. During the antenatal booking visit, body mass index (BMI) was estimated as a function of weight in kilograms per height in meter square. BMI was categorised by WHO classification: Underweight<18.5 kg/m$^2$, Normal (healthy weight) = 18.5–24.9 kg/m$^2$, overweight = 25.0–29.9 kg/m$^2$, obesity$\geq$ 30 kg/m$^2$ [34]. The early pregnancy BMI obtained at booking was used as a proxy for prepregnancy BMI.

**Exposure variables.** These included maternal age ($<$ 35 years, $\geq$ 35 years), parity (nulliparous, 1–3, $\geq$ 4), marital status (single or married), level of education (primary, secondary or tertiary), employment status (employed, unemployed), religion (Christianity, Islam), ethnicity (Yoruba versus non Yoruba), average monthly income level ($<$ 20,000, 20,000–99,999; $\geq$ 100,000), Lifestyle characteristics such as alcohol consumption (yes or no), and tobacco exposure (yes or no), BMI, SSB intake (low/high). The participants' level of physical activity is assesses using the (i.) passive transport (ownership of motorized transport) i.e. physically inactive while active transport or walking as physically active) [35,36] (ii.) Duration of moderate-intensity exercise (minutes per week) is the total time spent on moderate-intensity activity per week [37]. (iii.) Sedentary behaviour was also assessed using the pregnancy physical activity questionnaire (PPAQ) and classified as high or low according to the PPAQ instruction guide [38,39]. Psychological factors such as antepartum depression ascertained using the Edinburgh Postnatal Depression Scale (EPDS) score $\geq$ 12 [40]. Health care utilization assessed by the number of ANC visits ($<$ 4 or $\geq$4) as well as a history of chronic medical illness (yes or no).

## Pregnancy outcomes

The pregnancy outcomes assessed included gestational diabetes mellitus (GDM), mode of delivery, birth weight, gestational age at birth and birth asphyxia at 1 and 5 minutes. *Gestational Diabetes Mellitus*: GDM diagnosis was made based on the International Association of Diabetic and Pregnancy Study Group (IADPSG) criteria in which GDM was present if one of the thresholds FPG $\geq$ 5.1 mmol/l; 1-hour plasma glucose $\geq$ 10.0 mmol/l, 1-hour plasma glucose $\geq$ 8.5 mmol/l was surpassed. GDM was diagnosed based on a 75-g 2-hour oral glucose tolerance test (OGTT) at pregnancy 24–28 weeks [19]. *Mode of delivery* included spontaneous vaginal delivery (SVD), Caesarean section (CS), and Induction of labour (IOL). *Birth weight*: The birth weight of the infant at birth is grouped into Low Birth Weight ($<$ 2500g), Normal weight (2500 – 3999g) Macrosomia ($\geq$ 4000g) [41]. *Gestational age at delivery*: This included Preterm ($<$ 37 weeks), Term (37–42 weeks) Postdate ($>$ 42 weeks) [42]. *APGAR score*: Good ($\geq$ 7) or Birth asphyxia ($<$ 7) [43]. v.) *Postpartum Haemorrhage*: Blood loss $\geq$ 500 MLS post-vaginal delivery and $\geq$ 1000mls post-caesarean section [42,44].

## Statistical analyses

Statistical analyses were performed using STATA version 13 software for Windows (Stata Corp TX). The study participants were divided according to the category of GWG: insufficient, adequate, and excessive. The characteristics of pregnant women, namely sociodemographic, lifestyle, psychosocial, and maternal care utilisation, were assessed across the categories of GWG using the chi-square test. The dependent variables were insufficient and excessive GWG (adequate GWG was the reference category) and pregnancy outcomes. The factors associated with insufficient and excessive GWG were assessed using binary logistic regression, and we reported the Odds Ratio and 95% confidence interval. The factors investigated were maternal age, employment status, level of education, ethnicity, religion, monthly income, parity, body mass index, SSB intake, physical activity levels, duration of sleep, tobacco exposure, and alcohol consumption during pregnancy, depression, history of chronic medical disease, and the

number of antenatal care visit. Variables significant at a 5% level of statistical significance at binary logistic analysis were subjected to multiple logistic regression analyses (religion, income, BMI, physical activity level, and APD). The estimated unadjusted and adjusted odds ratios, 95% confidence intervals and p-values ($p < 0.05$) of associated factors were reported.

The incidence proportion of pregnancy outcomes by GWG was also assessed and reported. Poisson regression analysis was used to determine the association between GWG and pregnancy outcomes. Incidence risk ratio (relative risk); 95% CI and p-values were reported. Forest plots showing the association of specific pregnancy outcomes with GWG based on maternal BMI" were also presented.

## Results

### Participants' characteristics according to GWG

The characteristics of the study participants according to the GWG are shown in Table 1. The mean age of study participants was 29.8 ± 5.3 years and majority belonged to the Yoruba ethnic group (88.8%). The prevalence of GWG across the three categories were insufficient 337 (26.9%), adequate 211 (16.8%), and excessive 706 (56.3%) GWG. GWG was associated with maternal education ($p < 0.001$), religion ($p < 0.001$), monthly income ($p < 0.001$), physical activity ($p = 0.005$), BMI ($p < 0.001$), and antepartum depression ($p < 0.001$). Specifically, the level of education and income had a positive association with excessive GWG i.e. women with primary education or less: 15 (45.5%), secondary 151 (44.0%), and tertiary 537 (61.4%) and income: < *20,000*–192 (46.7%), *20,000–99,999*, 373 (63.2%), and ≥ *100,000*, 48 (61.5%). Also, more Christians (60.7%) than Muslims (49.6%) experienced excessive GWG. Women excessive GWG spent significantly less time on moderate intensity activity per week (25.3±23.1) minutes, more physically inactive (60.7%) and more sedentary (61.0) than women with inadequate GWG who spent (28.4 ± 23.6) minutes/week on moderate intensity activity, less more physically inactive (52.0%) and less sedentary (52.3%). Conversely, women who experienced antepartum depression (APD) had a higher proportion of insufficient GWG (39.2%) compared to women without APD (24.9%).

### Gestational weight by maternal body mass index

The pattern of GWG across maternal BMI is shown in Fig 2. The distribution of maternal BMI among the participants with GWG *assessment* was underweight 36 (3.0%), normal weight 622 (50.7%), overweight 337 (27.5%), and obese 232 (18.9%). Excessive GWG was the most familiar pattern of weight gain among the study participants across maternal BMI: underweight (55.6%), normal weight (47.3%), overweight (66.8%), and obese (60.4%). We observed adequate GWG was the least common.

### Factors associated with insufficient and excessive GWG among pregnant women in Ibadan, Nigeria

The factors associated with insufficient and excessive GWG are represented in Table 2. After adjusting for confounders, the factors related to GWG were maternal BMI, religion, income, level of physical activity and APD. Specifically, the odds of excessive GWG was associated with increasing revenue' 20,000–99,999' (AOR: 1.64, 95% CI: 1.25–2.17), '≥ 100,000' (AOR: 1.54, 95 CI: 0.90–2.64), compared to those who earned less than 20,000. Compared with women with normal weight, women with overweight (AOR: 2.07, 95% CI 1.52–2.88) and obese (AOR: 1.56, 95% CI 1.11–2.20) had a higher odds of excessive weight gain. Also women who had depression during pregnancy were also less likely to experience excessive GWG (AOR: 0.56,

**Table 1. Characteristics of pregnant women by GWG in Ibadan, Nigeria.**

| | Total (N-1254) | GWG | | | p-value |
| --- | --- | --- | --- | --- | --- |
| | | Insufficient (n = 337) | Adequate (n = 211) | Excessive (n = 706) | |
| **Proportion** | | **26.9%** | **16.8%** | **56.3%** | |
| **Age group** | | | | | |
| < 35 | 1004 (80.1) | 267(26.6) | 176(17.5) | 561(55.9) | 0.409 |
| ≥ 35 years | 250 (19.9) | 70(28.0) | 35 (14.0) | 145(58.0) | |
| **Parity** | | | | | |
| Nulliparous | 552(44.3) | 134 (24.3) | 102 (18.5) | 316 (57.2) | 0.126 |
| 1–3 | 636 (51.0) | 180 (28.3) | 102 (16.0) | 354 (55.7) | |
| ≥ 4 | 59 (4.7) | 22 (37.3) | 6 (10.2) | 31 (52.5) | |
| **Marital Status** | | | | | |
| Single | 67 (5.3) | 24 (35.8) | 6 (9.0) | 37 (55.2) | 0.094 |
| Married | 1187 (94.7) | 313 (26.4) | 205 (17.3) | 669 (56.3) | |
| **Maternal Education** | | | | | |
| At least Primary | 33 (2.6) | 13 (39.4) | 5 (15.1) | 15 (45.5) | **<0.001** |
| Secondary | 343 (27.5) | 128 (37.3) | 64 (18.7) | 151 (44.0) | |
| Tertiary | 894 (69.9) | 196 (22.4) | 141 (16.1) | 537 (61.5) | |
| **Employment Status** | | | | | |
| Employed | 1114 (88.8) | 307 (27.6) | 186 (16.7) | 621 (55.7) | 0.304 |
| Unemployed | 140 (11.2) | 30 (21.4) | 25 (17.9) | 85 (60.7) | |
| **Religion** | | | | | |
| Christianity | 733 (58.9) | 159 (21.7) | 129 (17.6) | 445 (60.7) | **<0.001** |
| Islam | 512 (41.1) | 177 (34.6) | 81 (15.8) | 254 (49.6) | |
| **Ethnicity** | | | | | |
| Yorubas | 1112 (88.8) | 311 (28.0) | 191 (17.2) | 610 (54.8) | 0.050 |
| Non-Yorubas | 140 (11.2) | 26 (18.6) | 20 (14.3) | 94 (67.1) | |
| **The income per month (Naira)*** | | | | | |
| <20,000 | 419 (38.1) | 148(36.0) | 71(17.3) | 192(46.7) | **<0.001** |
| 20,000–99,999 | 604 (54.9) | 134(22.1) | 98(16.2) | 373(63.2) | |
| ≥ 100,000 | 78 (7.1) | 14 (18.0) | 16 (20.5) | 48(61.5) | |
| *Lifestyle characteristics* | | | | | |
| **BMI** | | | | | |
| Underweight | 36 (2.9) | 9 (25.0) | 7 (19.4) | 20 (55.6) | **<0.001** |
| Normal weight | 622 (30.7) | 189 (30.4) | 139 (22.3) | 294 (47.3) | |
| Overweight | 337 (27.5) | 74 (22.0) | 38 (11.2) | 225 (66.8) | |
| Obese | 232 (18.9) | 65 (28.0) | 27 (11.6) | 140 (60.4) | |
| **SSB Intake** | | | | | |
| Low | 605 (49.8) | 151 (25.0) | 96(15.9) | 358(59.2) | 0.153 |
| High | 611 (50.2) | 175 (28.0) | 108(17.7) | 328(53.7) | |
| **Physical Activity** | | | | | |
| Physically active | 635(50.6) | 183(28.8) | 121(19.2) | 330(52.0) | 0.005 |
| Physically inactive | 619(49.4) | 154(24.8) | 89(14.5) | 376(60.7) | |
| **Sedentary Behaviour** | | | | | |
| Low | 561 (52.9) | 171 (30.5) | 95 (16.9) | 295 (52.3) | 0.005 |
| High | 518 (48.1) | 114 (22.0) | 88 (17.0) | 316 (61.0) | |
| **Moderate intensity activity Duration (minutes)** | 25.8±22.7 | 28.4±23.6 | 23.5±19.6 | 25.3±23.1 | 0.033 |
| **Sleep Duration (hours)** | 8.00±1.69 | 8.13± 1.64 | 8.07± 1.83 | 7.91± 1.66 | 0.114 |

*(Continued)*

**Table 1.** (Continued)

|  | Total (N-1254) | GWG | | | p-value |
|---|---|---|---|---|---|
|  |  | Insufficient (n = 337) | Adequate (n = 211) | Excessive (n = 706) |  |
| **Proportion** |  | **26.9%** | **16.8%** | **56.3%** |  |
| **Tobacco exposure** |  |  |  |  |  |
| Yes | 42 (3.4) | 11 (26.2) | 6 (14.3) | 25 (59.5) | 0.088 |
| No | 1212 (96.6) | 326 (26.9) | 205 (16.9) | 681 (56.2) |  |
| **Alcohol Consumption** |  |  |  |  |  |
| Yes | 164 (13.1) | 40 (24.4) | 28 (17.1) | 96 (59.5) | 0.738 |
| No | 1090 (86.9) | 297 (27.2) | 183 (16.8) | 610 (56.0) |  |
| *Psychosocial factors* |  |  |  |  |  |
| **Antepartum depression** |  |  |  |  |  |
| Yes | 176 (14.6) | 69 (39.2) | 29 (16.5) | 78 (44.3) | **<0.001** |
| No | 1025 (85.4) | 255 (24.9) | 173 (16.9) | 597 (58.2) |  |
| *Health care Utilisation* |  |  |  |  |  |
| **ANC visit** |  |  |  |  |  |
| <4visit | 198 (25.1) | 50 (25.2) | 33 (16.7) | 115 (58.1) | 0.684 |
| ≥ 4 visits | 590 (74.9) | 133 (22.5) | 95 (16.1) | 362 (61.4) |  |
| **Chronic medical disease** |  |  |  |  |  |
| Yes | 133 (10.6) | 42 (31.6) | 22 (16.5) | 69 (51.9) | 0.417 |
| No | 1121 (89.4) | 295 (26.3) | 189 (16.9) | 637 (56.8) |  |

*US Dollars equivalent < 20,000 (45 USD), 99,999 (220 USD), ≥ 100,000 (225 USD).

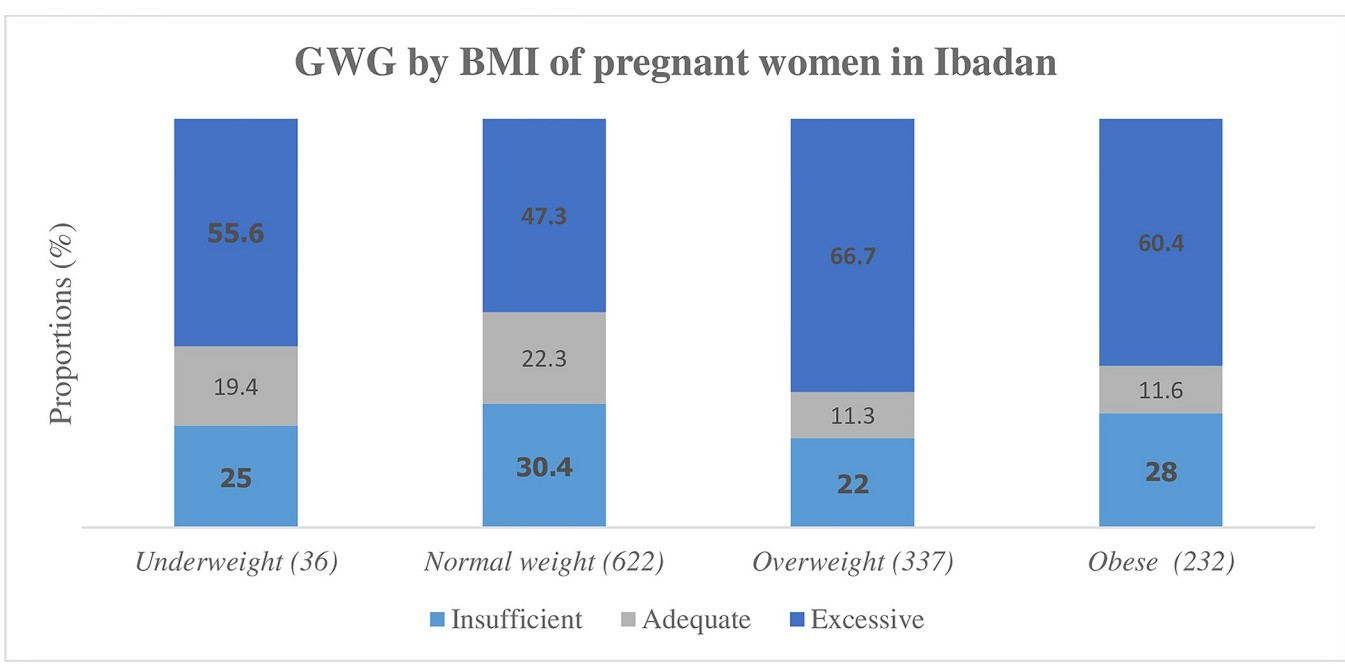

**Fig 2. GWG according to BMI in Ibadan, Nigeria.**

**Table 2.  Factors associated with insufficient and excessive GWG among pregnant women in Ibadan.**

| | Insufficient GWG | | Excessive GWG | |
|---|---|---|---|---|
| | Unadjusted OR (95%CI) | Adjusted OR (95% CI) | Unadjusted OR (95% CI) | Adjusted OR (95% CI) |
| **Age group** | | | | |
| Less than 35 | 1.00 | | 1.00 | |
| ≥ 35 years | 1.07 (0.79–1.46) | | 1.09 (0.82–1.44) | |
| **Employment** | | | | |
| Unemployed | 1.00 | | 1.00 | |
| Employed | 1.40 (0.91–2.13) | | 0.82 (0.57–1.17) | |
| **Education** | | | | |
| ≤ Primary | 1.00 | 1.00 | 1.00 | |
| Secondary | 0.92 (0.44–1.90) | 1.13(0.49–2.55) | 0.94 (0.46–1.93) | |
| Tertiary | 0.45 (0.22–0.91)** | 0.78 (0.35–1.73) | 1.91 (0.95–3.85) | |
| **Ethnicity** | | | | |
| Non Yorubas | 1.00 | | 1.00 | |
| Yorubas | 1.70 (1.09–2.66) | | 0.60 (0.41–0.86) | |
| **Religion** | | | | |
| Christian | 1.00 | 1.00 | 1.00 | 1.00 |
| Islam | 1.91 (1.48–2.46)** | 1.52 (1.12–2.08)** | 0.64 (0.51–0.80)** | 0.73 (0.55–0.97)** |
| **Income** | | | | |
| < 20,000 | 1.00 | 1.00 | 1.00 | 1.00 |
| 20,000–99,999 | 0.51 (0.39–0.68)** | 0.53(0.41–0.76)** | 1.90 (1.47–2.44)** | 1.76 (1.31–2.37)** |
| ≥ 100,000 | 0.39 (0.21–0.72)** | 0.36 (0.17–0.74)** | 1.89 (1.15–3.10)** | 1.51 (0.85–2.70) |
| **Parity** | | | | |
| Nulliparous | 1.00 | | 1.00 | |
| 1–3 | 1.23 (0.95–1.60) | | 0.94 (0.75–1.18) | |
| ≥ 4 | 1.86 (1.06–3.26) | | 0.83 (0.48–1.42) | |
| **BMI** | | | | |
| Underweight | 0.76 (0.35–1.66) | 0.53 (0.21–1.38) | 1.40 (0.71–2.74) | 2.02 (0.89–4.53) |
| Normal weight | 1.00 | 1.00 | 1 | 1 |
| Overweight | 0.65 (0.47–0.88)** | 0.69 (0.48–0.99)** | 2.07 (1.52–2.88)** | 2.12 (1.52–2.95)** |
| Obese | 0.89 (0.64–1.25) | 0.98 (0.66–1.45) | 1.70 (1.25–2.31)** | 1.47 (1.02–2.13)** |
| **SSB Intake** | | | | |
| Low | 1.00 | | 1.00 | 1.00 |
| High | 1.20 (0.93–1.56) | | 0.80 (0.63–1.00)* | 0.78 (0.57–1.06) |
| **Physical Activity** | | | | |
| Physically active | 1.00 | | 1.00 | 1.00 |
| Physically inactive | 0.82(0.63–1.06) | | 143(1.14–1.79)** | 1.25 (0.94–1.65) |
| **Sedentary Behaviour** | | | | |
| Low | 1.00 | | 1 | |
| High | 0.64 (0.49–0.85)** | 0.75 (0.55–1.02) | 1.41 (1.11–1.80)** | 1.26(0.96–1.67) |
| **Moderate intensity activity Duration (minutes)** | 1.07 (0.99–1.12) | | 0.99 (0.99–1.002) | |
| **Sleep Duration** | 1.01 (1.001–1.01 | | 0.93 (0.87–0.99)** | 0.95 (0.88–1.04) |
| **Tobacco exposure** | | | | |
| Yes | 0.96 (0.48–1.94) | | 1.15 (0.61–2.15) | - |
| No | 1.00 | | 1.00 | - |
| **Alcohol Consumption** | | | | |
| Yes | 0.86 (0.59–1.26) | | 1.11 (0.80–1.55) | - |
| No | 1.00 | | 1.00 | - |

*(Continued)*

**Table 2.** (Continued)

|  | Insufficient GWG | | Excessive GWG | |
|---|---|---|---|---|
|  | Unadjusted OR (95%CI) | Adjusted OR (95% CI) | Unadjusted OR (95% CI) | Adjusted OR (95% CI) |
| **Antepartum depression** |  |  |  |  |
| Yes | 1.95 (1.39–2.72)** | 1.79 (1.21–2.65)** | 0.57 (0.41–0.79)** | 0.53 (0.36–0.79)** |
| No | 1.00 | 1.00 | 1.00 |  |
| **ANC visits** |  |  |  |  |
| <4visit | 1.00 |  | 1.00 | - |
| ≥ 4 visits | 0.86 (0.59–1.25) |  | 1.15 (0.83–1.59) | - |
| **Chronic medical condition** |  |  |  |  |
| Yes | 1.29 (0.88–1.91) |  | 0.82 (0.57–1.17) | - |
| No | 1.00 |  | 1.00 | - |

95% CI 0.36–0.82). Conversely, compared with Christian women, Muslims had higher odds for insufficient GWG (AOR: 1.52, 95% CI 1.13–2.05). And women with APD had 70% higher odds of insufficient GWG (AOR: 1.70, 95% CI 1.17–2.45) than women without APD.

## Pregnancy outcomes associated with GWG

The incidence proportions, relative risk and 95% CI of the pregnancy outcomes are shown in Table 3. The incidence of caesarean section increased with GWG: insufficient GWG (25.3%), adequate (29.3%): an excessive (36.7%). Also, women with excessive GWG had a 26% higher risk of undergoing a caesarean section: [unadjusted RR 1.26 (95% CI) (0.94–1.68); p-value: 0.129] compared to adequate GWG. Conversely, the incidence of SVD decreased with GWG, with excessive GWG having a 24% lower risk of SVD: [unadjusted RR 0.86 (95% CI) (0.70–1.05); p value< 0.145]. Excessive GWG was also associated with an increased risk of postpartum haemorrhage [unadjusted RR 1.24 (95% CI) (0.81–1.89); p value< 0.318]. There were no statistically significant associations observed between pregnancy outcomes and gestational weight gain.

Forest plots showing the association of inadequate and excessive GWG with pregnancy outcomes (caesarean section, spontaneous vaginal delivery, macrosomia and postpartum haemorrhage) stratified by BMI are shown in Fig 3.

## Discussion

GWG is a neglected maternal health issue in Nigeria with a high maternal and neonatal health burden. Appropriate GWG is crucial for achieving optimal perinatal outcomes and preventing postpartum weight retention, future obesity and cardiometabolic complications. Hence optimal GWG contributes to health, well-being and sustainable development in women and children. Using the Ibadan Pregnancy Cohort study, we investigated the pattern, prevalence, predictors and outcomes of GWG among pregnant women in Ibadan. In this study, we assessed GWG using the rate of weight gain per week rather than the total weight gain, a more precise measure, This was because information required for the estimation total weight gain i.e. the weight just before the onset of labour and the pre-pregnancy weight, are not available in our study setting. Hence, in such a situation IOM guidelines recommends rate of weight gain per week assessed within the two and third trimester [2,4,8] which has also been put to use in other settings [4,8,6,45].

Importantly, we found that only 16.8% of our respondents achieved normal GWG, 26.9% had insufficient GWG, and more than half (56.3%) had excessive GWG. Excessive weight gain

**Table 3. Association between GWG and pregnancy outcomes in Ibadan, Nigeria.**

| Pregnancy Outcomes | n/N | Incidence (%) | Unadjusted Relative Risk 95% CI | P-value |
|---|---|---|---|---|
| **Caesarean section** | | | | |
| *Insufficient* | 74/234 | 25.3 | 0.87 (0.61–1.23) | 0.420 |
| *Adequate* | 55/188 | 29.3 | 1.00 | |
| *Excessive* | 234/637 | 36.7 | 1.26 (0.94–1.68) | 0.129 |
| **Spontaneous vaginal delivery** | | | | |
| *Insufficient* | 206/293 | 70.3 | 1.07 (0.85–1.33) | 0.589 |
| *Adequate* | 127/192 | 66.2 | 1.00 | |
| *Excessive* | 366/643 | 56.9 | 0.86 (0.70–1.05) | 0.145 |
| **Induction of Labour** | | | | |
| *Insufficient* | 5/293 | 1.7 | 1.40 (0.13–1.25) | 0.117 |
| *Adequate* | 8/192 | 4.2 | 1.00 | |
| *Excessive* | 29/643 | 4.5 | 1.08 (0.49–2.37) | 0.843 |
| **Macrosomia** | | | | |
| *Insufficient* | 14/236 | 5.6 | 1.39 (0.56–3.44) | 0.475 |
| *Adequate* | 7/174 | 4.0 | 1.00 | |
| *Excessive* | 36/589 | 6.1 | 1.51 (0.68–3.41) | 0.311 |
| **Low birth weight** | | | | |
| *Insufficient* | 16/250 | 6.2 | 1.01 (0.47–2.18) | |
| *Adequate* | 11/174 | 6.3 | 1.00 | |
| *Excessive* | 48/589 | 8.2 | 1.28 (0.68–2.48) | 0.447 |
| **Preterm delivery** | | | | |
| *Insufficient* | 43/272 | 15.8 | 1.17 (0.71–1.93) | 0.532 |
| *Adequate* | 24/178 | 13.5 | 1.00 | |
| *Excessive* | 82/609 | 13.5 | 0.99 (0.63–157) | 0.995 |
| **Birth Asphyxia at 1 minute** | | | | |
| *Insufficient* | 22/165 | 13.3 | 0.71 (0.39–1.27) | 0.248 |
| *Adequate* | 24/128 | 18.8 | 1.00 | |
| *Excessive* | 80/476 | 16.8 | 0.90(0.57–1.41) | 0.638 |
| **GDM** | | | | |
| *Insufficient* | 29/155 | 18.7 | 1.09 (0.62–1.91) | 6.771 |
| *Adequate* | 21/122 | 17.2 | 1.00 | |
| *Excessive* | 73/340 | 21.5 | 1.25 (0.77–2.02) | 0.372 |
| **Postpartum Haemorrhage** | | | | |
| *Insufficient* | 33/293 | 11.3 | 0.80 (0.48–1.33) | 0.392 |
| *Adequate* | 27/192 | 14.1 | 1.00 | |
| *Excessive* | 112/643 | 17.4 | 1.24 (0.81–1.89) | 0.318 |

was more prevalent in our study population than insufficient GWG, a pattern commonly reported in high and middle-income countries like Canada, Australia and Brazil [15,16,46,47]. Conversely, low and middle-income countries have reported a higher prevalence of inadequate GWG than excessive [13,15,16,47–49]. Excessive GWG is an emerging maternal health issue in transiting economies with increased adverse perinatal outcomes such as macrosomia, increased operative delivery, postpartum weight retention, and maternal obesity in the subsequent pregnancies and future risk of non-communicable diseases [7,10,24,25,50–52]. The prevalence of excessive GWG reported in Nigeria ranges from 10.5–29.4% [30,53]. For example, Senbanjo et al. (2021) reported a lower prevalence (11.1%) among women in a tertiary

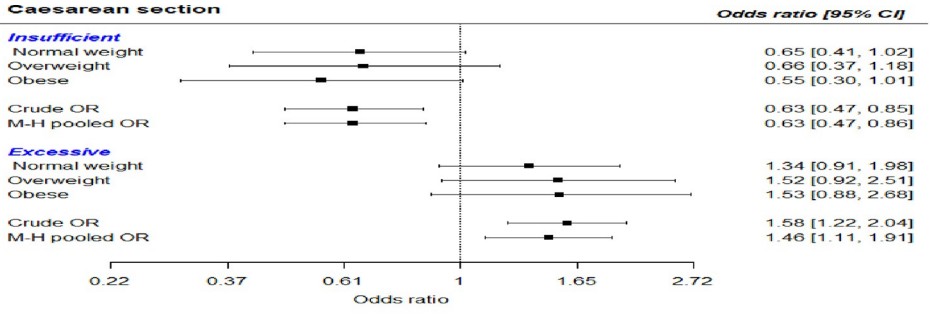

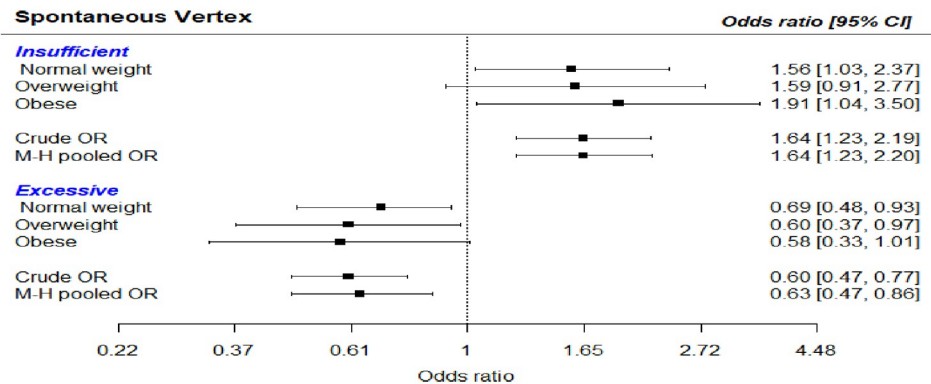

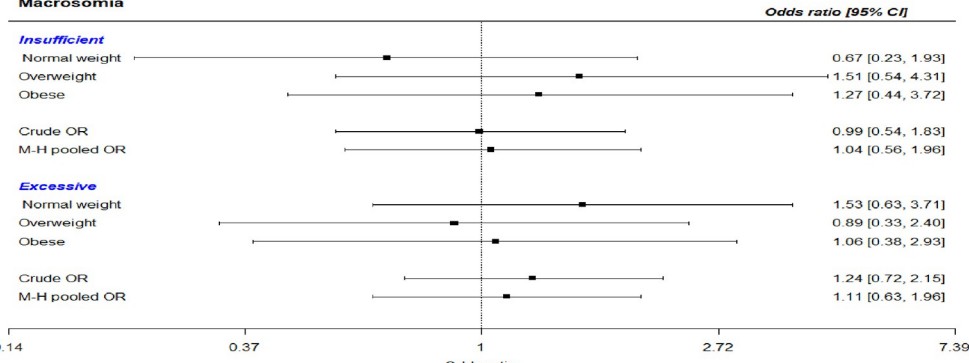

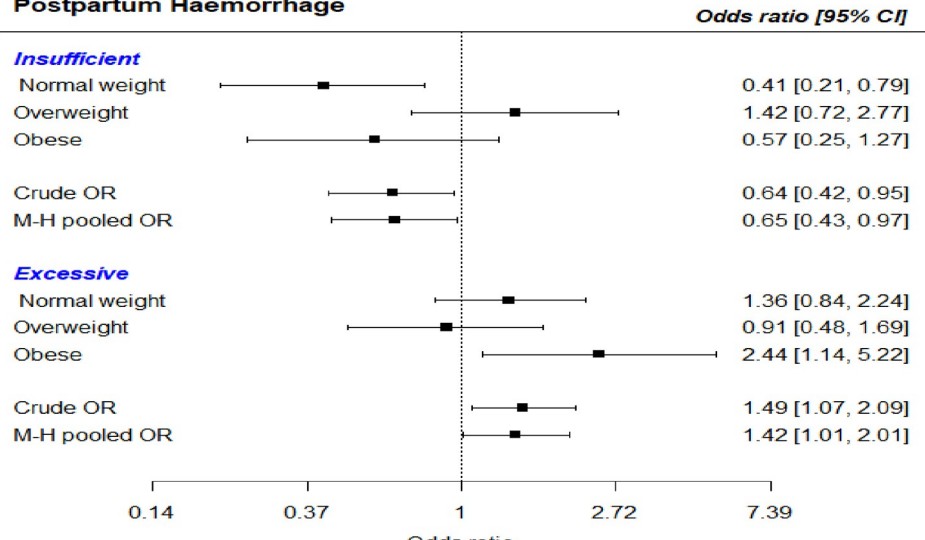

**Fig 3. 1_2_3_4: Forest plots showing the association of specific pregnancy outcomes with GWG by maternal BMI (caesarean section, spontaneous vaginal delivery, macrosomia and postpartum haemorrhage).** Footnote for the forest plots [1.] Underweight weight was absent from the forest plots because they made up a very low proportion of the final analysis 36 (2.9%) and was mostly omitted from the output of the stratified analysis for the forest plots.

health facility in Lagos, Nigeria. The variation in prevalence may be due to the differences in the participants' characteristics, including the sociodemographic and lifestyle factors, timing of the study, study design, built environment and stressors in their study setting [54]. However, these researchers also reported a much higher prevalence of 54.1% for insufficient GWG. A recent systematic review and meta-analysis (2020) on the pattern of GWG in sub-Saharan Africa documented that insufficient GWG was the most common form of GWG occurring in more than half of the population [19]. Nonetheless, they also observed that excessive GWG was predominant in rich African countries such as South Africa, Nigeria, Ghana and Kenya [19].

Notably, the health systems in sub-Saharan Africa have focussed on maternal undernutrition and insufficient GWG from inadequate food intake, poor dietary quality, recurrent infections and short inter-pregnancy intervals more frequently than excessive weight gain and its complications [55]. Therefore, Nigeria's maternal health care system should implement interventions that promote appropriate GWG. These interventions encompass creating awareness on the appropriate gestational weight gain (GWG) range, encouraging a healthy lifestyle during pregnancy through healthy diet and physical activity, routine self-monitoring of weight, preventing infection, and providing information on healthy weight gain and monitoring GWG among pregnant women [48].

Investigating the risk factors associated with inappropriate GWG is essential for providing targeted interventions for addressing the factors that contribute to undesirable weight gain during pregnancy among Nigerian women. We examined various factors associated with GWG, including sociodemographic, lifestyle, psychological, and healthcare factors. However, after adjusting for confounders, we found maternal BMI, religion, income, and APD significant factors. High income had a positive relationship (AOR = 1.68) with excessive GWG, with women with higher earnings having a higher odds of gaining excess weight. The positive association between income or high socioeconomic status and weight gain, particularly in LMIC developing countries, have been established in the literature [56,57]. High income affords more significant resources for improved access to food and increased food intake, including processed foods and sweetened drinks.

Additionally, undue reliance on technological and automated devices promotes physical inactivity and sedentary behaviours, which boosts weight gain. Conversely, high socioeconomic status is associated with an inverse relationship with weight gain in high-income countries because of the increased health literacy of the complications related to excessive weight gain [56–58]. Paul et al. (2013) corroborated our finding by reporting that low-income earning women in the US gained more weight than high-income earning women because they engaged in more weight-promoting behaviours [59].

We also found that women who were overweight or obese had a higher odds of excessive GWG than women with normal weight. This finding contradicts the 2009 IOM GWG guideline, which recommends a narrow weight gain limit for women with obesity [2] to minimise the adverse pregnancy outcomes associated with adiposity [13]. This finding could imply a lack of awareness of IOM guidelines on GWG in our maternal care services. A Nigerian study corroborated our study by reporting that more than half of their respondents said a lack of awareness of the risk involved with inappropriate GWG. At the same time, 22.4% of respondents reported that their healthcare provider had recommended that the women not gain

weight beyond a specific limit (12–12.5kg)—not based on the IOM guidelines [31]. The IOM guidelines are not yet in place, nor is there any consensus on GWG in Nigeria's maternal care [27,31]. For instance, based on the IOM guidelines, the American College of Obstetricians and Gynaecologists recommends that clinicians calculate a woman's prepregnancy BMI at the first antenatal care visit and educate her on the importance of appropriate GWG goals [28]. Adopting and implementing a GWG guideline in Nigeria's maternal health services is essential. Although the IOM guideline is based on cut-off points from white women and is yet to be validated among Africans, the IOM guidelines are still the most utilised GWG guideline. Hence it could be helpful in Nigeria up until local guidelines are developed.

Antenatal care provides a platform for education, counselling and monitoring GWG and the short and long-term problems associated with inappropriate weight gain should be discussed during ANC visits. In our study, the number of antenatal care attendance did not influence GWG. Conversely, compared with Christians, Muslims had higher odds for insufficient GWG (AOR: 1.52). Perhaps, because the religion imposes certain food restrictions, for example, on the pork intake that Christians are free to eat. The Ramadan fast may also prevent excessive weight gain because of reduced food intake. A Turkish study demonstrated a low prevalence of excessive GWG, probably for the same reasons [17]. Also, women with APD had 70% higher odds of insufficient GWG than women without APD. The reduced appetite, weight loss, poor self-care practices, and inadequate nutrition experienced by women with APD could explain this finding [60,61].

Importantly, our study investigated the influence of maternal lifestyle characteristics–SSB intake, sleep pattern and physical on GWG among our study population. We assessed physical activity using active/passive transport [35], duration of moderate-intensity exercise [35,37] and sedentary behaviour [38,39]. Active transport is a measure of physical activity because it contributes to total physical activity and increases energy expenditure [35,36]. Women excessive GWG spent less time on moderate intensity activity per week minutes, more physically inactive and more sedentary than women with inadequate GWG. We found that physically in active and sedentary women had higher odds for excessive GWG than physically active women. Although, the relationship became insignificant on multivariate analysis. Importantly, the WHO recommends that pregnant women engage in at least 150 minutes of moderate-intensity physical activity during the week [37]. Researchers have reported poor compliance with the recommendation among Nigerian pregnant women [62,63]. Hence the need to actively promote physical activity among pregnant women to obtain benefits, including improving cardiovascular fitness, preventing excessive GWG and GDM, improving sleep quality and so on [64–66]. Although there was a significant relationship between sleep duration and GWG, women with excessive GWG had a shorter sleep duration than those with adequate and insufficient sleep. Researchers have found an association between sleep deprivation and weight gain [67]. Further studies should further explore this relationship among the Nigerian pregnant population. In addition, although our study did not find any significant relationship between SSB intake and GWG, SSBs have been associated with obesity, weight gain and excessive GWG in the literature [68–70] because of the high and easily absorbed sugar content that leads to high energy intake and weight gain [71].

Notably, we estimated the incidence and relative risks of pregnancy outcomes by GWG. We found that GWG had an association with the mode of delivery. Excessive GWG had a 26% higher risk for the caesarean section (the association was the same for emergency or elective caesarean section). Although there was no statistical significance, this does not diminish its clinical or public health relevance. Studies have shown that excessive GWG increases the risk of caesarean delivery [23,72,73]. Plausible reasons for this association include a higher odds of maternal complications and cephalo-pelvic disproportion arising from the presence of a big

foetus. Excessive GWG is accompanied by excess metabolic fuels, i.e. high maternal glucose, free fatty acid, and amino acid concentrations that are passed to the growing foetus leading to increase foetal weight. Even though not statistically significant, excessive GWG was associated with a 51% and 24% higher risk of macrosomia and postpartum haemorrhage than women with adequate GWG. While some studies have shown a positive association between excessive weight gain and gestational diabetes mellitus, a few have shown no association [15,74,75]. The lack of association between GWG and GDM in our study is likely due to lifestyle interventions and weight control, which restrict weight gain among women with GDM who obtain maternal care from facilities that provide comprehensive care [76]. Maternal BMI and GWG synergise maternal outcomes as the coexistence of obesity and excessive GWG leads to more severe complications [8,50]. For instance, studies among Chinese women showed that obese women with excessive GWG have much higher odds for hypertensive disorders during pregnancy than normal-weight women with adequate weight gain [23–25]. We found that excessive weight gain significantly increased the risk of postpartum haemorrhage among women who were obese. The association between maternal obesity postpartum haemorrhage has been reported in the literature [42,77,78]. The plausible reasons include poor uterine contractility in obese women compared with non-obese [79], increase foetal weight that could lead to uterine atony or perineal tear, associate large placenta and placenta praevia [78]. Blomberg et al. in the cohort study among Swedish pregnancy reported that the risk of atonic uterine haemorrhage increased rapidly with BMI [80]. Therefore, extra vigilance is advised in the active management of the third stage of labour in women with obesity and excessive GWG [78,79].

This study contributes significantly to a critical gap in maternal health literature in Nigeria, in which GWG has been scantly examined. Our study has several strengths, including the use of a prospective cohort study design which allowed for the direct measurement of critical variables, especially serial maternal weights, obtained by trained staff during every antenatal visit. Notably, previous studies had primarily relied on retrospective studies or self-reports subject to recall or misclassification bias. The prospective study design also allowed the investigation of multiple risk factors and outcomes associated with GWG. We investigated the influence of a broader range of variables; sociodemographic, lifestyle (alcohol consumption, tobacco exposure, SSB intake, physical activity and sleep duration), healthcare utilisation and psychological factors such as APD, which were lacking in previous studies. The use of multiple health facilities also enhanced the generalizability of the study. However, our study also has limitations. First is the bias from losses to follow-up, which is typical of prospective cohort studies but was accounted for by assuming a 40% attrition in the sample size calculation. In this study, we could not estimate total GWG because prepregnancy and delivery weights were unavailable; hence, we used a weekly GWG rate. Also, the use of booking weight at $\leq$ 20 weeks (mean 16.2 ± 4.7 weeks) women would have experienced some weight gain could have potentially influenced gestational weight gain estimate. Therefore, GWG rate used in this study was a more appropriate measure as recommended by IOM guidelines [4].

## Conclusions

Excessive GWG was the commonest form of GWG in our study population, with a prevalence of 56.3%, but insufficient and adequate GWG were observed in 26.9% and 16.8% of the pregnant women, respectively. The significant predictors of GWG were income, maternal BMI, religion, physical activity and APD. Even though the incidences of caesarean section and postpartum haemorrhage were higher among women with excessive GWG compared with adequate and insufficient GWG, we did not find any significant association between excessive GWG and pregnancy outcomes. Maternal obesity modified the relationship between GWG

and pregnancy outcomes. Achieving appropriate GWG should be prioritise in Nigeria's maternal health care.

## Acknowledgments

Our gratitude goes to our research team for their dedication and hard work, the health workers at our study sites for their cooperation and support, and our study participants. Special thanks go to the Consortium for Advanced Research Training for Africa (CARTA) for the training, care, support.

## Author Contributions

**Conceptualization:** Ikeola A. Adeoye.

**Data curation:** Ikeola A. Adeoye.

**Formal analysis:** Ikeola A. Adeoye.

**Funding acquisition:** Ikeola A. Adeoye.

**Investigation:** Ikeola A. Adeoye.

**Methodology:** Ikeola A. Adeoye.

**Project administration:** Ikeola A. Adeoye.

**Resources:** Ikeola A. Adeoye.

**Software:** Ikeola A. Adeoye.

**Supervision:** Ikeola A. Adeoye, Elijah A. Bamgboye, Akinyinka O. Omigbodun.

**Validation:** Ikeola A. Adeoye, Elijah A. Bamgboye, Akinyinka O. Omigbodun.

**Visualization:** Ikeola A. Adeoye, Elijah A. Bamgboye, Akinyinka O. Omigbodun.

**Writing – original draft:** Ikeola A. Adeoye.

**Writing – review & editing:** Ikeola A. Adeoye, Elijah A. Bamgboye, Akinyinka O. Omigbodun.

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
