## [Decision Letter · Decision Letter 0]

16 Feb 2023

PONE-D-22-35461Gestational Weight Gain among pregnant women in Ibadan, Nigeria: pattern, predictors and pregnancy outcomes.PLOS ONE

Dear Dr. Adeoye,

Thank you for submitting your manuscript to PLOS ONE. After careful consideration, we feel that it has merit but does not fully meet PLOS ONE’s publication criteria as it currently stands. Therefore, we invite you to submit a revised version of the manuscript that addresses the points raised during the review process.

We look forward to receiving your revised manuscript.

Kind regards,

Malshani Lakshika Pathirathna, PhD

Academic Editor

PLOS ONE

Journal Requirements:

2. You indicated that you had ethical approval for your study. Please clarify whether minors (participants under the age of 18 years) were included in this study. If yes, in your Methods section, please ensure you have also stated whether you obtained consent from parents or guardians of the minors included in the study or whether the research ethics committee or IRB specifically waived the need for their consent.

"This research was supported by the Consortium for Advanced Research Training in Africa (CARTA). CARTA is jointly led by the African Population and Health Research Center and the University of the Witwatersrand and funded by the Carnegie Corporation of New York (Grant No. G-19-57145), Sida (Grant No:54100113), Uppsala Monitoring Center, Norwegian Agency for Development Cooperation (Norad), and by the Wellcome Trust [reference no. 107768/Z/15/Z] and the UK Foreign, Commonwealth & Development Office, with support from the Developing Excellence in Leadership, Training and Science in Africa (DELTAS Africa) programme. Ikeola Adeoye is a CARTA PhD fellow. The statements made and views expressed are solely the responsibility of the Fellow. The funders had no role in study design, data collection and analysis, decision to publish, or preparation of the manuscript."

Reviewers' comments:

Reviewer's Responses to Questions

**Comments to the Author**

1. Is the manuscript technically sound, and do the data support the conclusions?

Reviewer #1: Partly

Reviewer #2: Partly

2. Has the statistical analysis been performed appropriately and rigorously? 

Reviewer #1: Yes

Reviewer #2: No

3. Have the authors made all data underlying the findings in their manuscript fully available?

Reviewer #1: Yes

Reviewer #2: Yes

4. Is the manuscript presented in an intelligible fashion and written in standard English?

Reviewer #1: Yes

Reviewer #2: No

5. Review Comments to the Author

Reviewer #1: This study determines factors affecting gestational weight gain and its association with maternal outcomes in Nigeria. Please find below my individual comments.

Abstract

The results did not reflect all findings from the study, e.g., findings from Table 4

Introduction

1. Overall, the introduction section is slightly general without the support of facts or evidence, such as percentages or rates. This information is important to support the study's rationale and for the reader to understand the actual situation of problems. Suggest including some background information on the GWG pattern in Nigeria and other related regions

a. Line 124-129: Suggest elaborate these statements, e.g., What is the percentage of excessive and insufficient in the countries?

b. Line 129-131: please elaborate

Figure: Suggest replacing Figure 1 with the flow chart of study participants, whereby how the study participants were selected.

Materials and Methods:

2. Line 164: …at investing.., suggest …to investigate..

3. Line 215- 254: Since lifestyle information was collected in this study, especially the sugar-sweetened beverages, and sleep pattern, and physical activity (line 178-180), wondering why the authors did not investigate the associations between lifestyle factors and GWG. This will strengthen the value of the article. Suggest state clearly how the variables were selected to be included in the model

4. Line 256: please list out the adjusted variables

5. Line 209: please include the source of reference

Results:

6. Line 315: “'20,000-99,999', please provide a footnote on the rate of USD for ease of interpretation

Discussion:

7. Line 406: .. had appropriate weight.., suggest normal GWG

8. Line 423-430: Since the authors noticed that maternal lifestyle behaviours, such as nutrition and physical activity, are important factors contributing to GWG, suggesting the author included these factors in the analysis (Table 2)

9. Line 451-455: please include the source of reference; what is the recommended GWG practice in Nigeria? Suggest including this information in the introduction

10. Line 472-474: Any information on the types of cesarean sections performed? Elective or emergency cesarean section

Reviewer #2: I would like to thank the authors for conducting the study on this important and timely topic. please find my specific comments below.

1. Better to clearly report what type of logistic regression (binary or multinomial) was conducted to examine associations.

2. The authors defined abbreviations multiple times. For example, "Gestational weight gain (GWG)" was defined many times. Please revisit the use of abbreviations.

3. The author bluntly reported the last weight as "the third trimester weight", which is vague. They need to report the exact gestational week at which the last maternal weight measured.

4. They also stated that "The factors associated with insufficient and excessive GWG were assessed using bivariate logistic regression" What do they mean by "bivariate logistic regression?"

5. In table 3, the authors chose insufficient GWG as reference group while examining the association between GWG and pregnancy outcomes. Since insufficient and excessive GWG are undesirable outcomes, they are supposed to compare their outcomes with the outcomes of adequate GWG. I would suggest them to change the reference group from insufficient weight gain to adequate GWG and report the effect of insufficient and excessive GWG on pregnancy outcomes. I would also suggest them to report whether the finding in the table three are crude or adjusted. If the they have adjusted for confounders, they need to report the list of variables for which the analysis were adjusted. Similarly, the findings in table 4 were a bit confusing. No reference group. It is unclear what "-" mean.

6. It would have been good if the authors reported how many of the participants were underweight, normal weight, overweight and obese.

7. It would be good if the authors commented on the implication and appropriateness of using the IOM recommendations for GWG for Nigerian women.

8. Gestational weight gain was measured as the difference between the last third trimester weight recorded and the booking weight divided by the number of weeks between the weights. The authors need to substantiate the appropriateness of this GWG calculation method with evidence.

9. The gestational age for booking weight was 20 weeks. Does this time have any implication of GWG?

Thank you and good luck.

6. PLOS authors have the option to publish the peer review history of their article (what does this mean?). If published, this will include your full peer review and any attached files.

Reviewer #1: No

Reviewer #2: No

---

## [Author Response · Author response to Decision Letter 0]

25 Apr 2023

POINT-BY-POINT RESPONSE TO THE REVIEWER'S COMMENT

"Gestational Weight Gain among Pregnant women in Ibadan, Nigeria: pattern, predictors and pregnancy outcomes"

Reviewer #1: 

This study determines factors affecting gestational weight gain and its association with maternal outcomes in Nigeria. Please find below my comments. 

Abstract 

The results did not reflect all findings from the study, e.g., findings from Table 4 

 Table 4 has been removed from the manuscript

Introduction 

 Overall, the introduction section is slightly general without the support of facts or evidence, such as percentages or rates. This information is important to support the study's rationale and for the reader to understand the actual situation of problems. Suggest including some background information on the GWG pattern in Nigeria and other related regions 

 The introduction is re-written and supported with evidence

 Line 124-129: Suggest elaborate these statements, e.g., What is the percentage of excessive and insufficient in the countries?

 Done

 Line 129-131: please elaborate

 Done

Figure: Suggest replacing Figure 1 with the flow chart of study participants, whereby how the study participants were selected. 

 Figure 1 is now a flow chart of study participants

Materials and Methods: 

 Line 164: …at investing.., suggest …to investigate.

 Replaced with "to investigate."

 Line 215- 254: Since lifestyle information was collected in this study, especially the sugar-sweetened beverages, and sleep pattern, and physical activity (line 178-180), wondering why the authors did not investigate the associations between lifestyle factors and GWG. 

 Additional lifestyle factors added and presented in Tables 1 & 2

 sugar-sweetened beverage intake

 sleep pattern

 physical activity

Importantly, our study investigated the influence of maternal lifestyle characteristics – SSB intake, sleep pattern and physical on GWG among our study population. We found that less physically active women had higher odds for excessive GWG than more physically active women. The WHO recommends that pregnant women engage in at least 150 minutes of moderate-intensity physical activity during the week [50]. Researchers have reported poor compliance with the recommendation among Nigerian pregnant women [51, 52]. Hence the need to actively promote physical activity among pregnant women to obtain benefits, including improving cardiovascular fitness, preventing excessive GWG and GDM, improving sleep quality and so on [53-55]. Although there was a significant relationship between sleep duration and GWG, women with excessive GWG had a shorter sleep duration than those with adequate and insufficient sleep. Researchers have found an association between sleep deprivation and weight gain. Further studies should further explore this relationship among the Nigerian pregnant population.

How the variables were selected to be included in the model

 Variables significant at a 5% level of statistical significance at binary logistic analysis were subjected to multiple logistic regression analyses (religion, income, BMI, physical activity level, and APD). The estimated unadjusted and adjusted odds ratios, 95% confidence intervals and p-values (p<0.05) of associated factors were reported. 

Line 256: please list out the adjusted variables 

 Done

 Line 209: please include the source of reference 

 Done

Results:

 Line 315: "'20,000-99,999', please provide a footnote on the rate of USD for ease of interpretation 

 a footnote on the rate of USD provided in Table 1

Discussion:

Line 406: had appropriate weight.., suggest normal GWG 

 Replaced with Normal GWG

Line 423-430: Since the authors noticed that maternal lifestyle behaviours, such as nutrition and physical activity, are important factors contributing to GWG, suggesting the author included these factors in the analysis (Table 2) 

 Added to the analysis

Line 451- 455: please include the source of reference; what is the recommended GWG practice in Nigeria? Suggest including this information in the introduction 10. 

Line 472-474: Any information on the types of cesarean sections performed? Elective or emergency cesarean section 

 The direction and association with GWG was similar for All cesarean sections, Elective or emergency cesarean section 

Reviewer #2

I would like to thank the authors for conducting the study on this important and timely topic. Please find my specific comments below.

 Better to clearly report what type of logistic regression (binary or multinomial) was conducted to examine associations.

 Binary logistic regression performed and reported in the manuscript

 The authors defined abbreviations multiple times. For example, "Gestational weight gain (GWG)" was defined many times. Please revisit the use of abbreviations.

 use of abbreviations revisited Gestational weight gain (GWG) now defined once 

 The author bluntly reported the last weight as "the third trimester weight", which is vague. They need to report the exact gestational week at which the last maternal weight measured.

 .According to the literature there are three ways of estimating GWG 

 Total weight gain – This is the difference the woman’s weight just before the onset of labour and her pre-pregnancy weight. (These pieces of information are not readily available in settings with poor medical record keeping, lack of awareness of health workers on the importance of GWG, high out of hospital delivery. This situation is prevalent in several hospitals. 

 Net weight gain – This is the difference the woman’s total weight gain and birthweight of the infant

 Rate of weight gain per week – This is a measure weight gain over a particular time period in weeks divided by the time interval in weeks. This rate could be assessed within the 2nd and third trimesters. This measure would appeal to researchers from developing countries, which was the measure used in this study, where GWG rate as assessed as the difference between the final weight recorded by the woman during antenatal care in the third trimester and the weight at the booking visit.

 In this study we used the rate of weight gain per week because weight at delivery was not assessible

 That is the final third-trimester weight minus booking weight divided by gestational age, at which the final third-trimester weight was measured minus gestational age at booking. 

GWG rate = (third trimester weight (final)- weight at booking)/(GA at third trimester weight (final)- GA at booking)

4. They also stated that "The factors associated with insufficient and excessive GWG were assessed using bivariate logistic regression" What do they mean by "bivariate logistic regression?" 

 binary logistic regression

 In table 3, the authors chose insufficient GWG as reference group while examining the association between GWG and pregnancy outcomes. Since insufficient and excessive GWG are undesirable outcomes, they are supposed to compare their outcomes with the outcomes of adequate GWG. I would suggest them to change the reference group from insufficient weight gain to adequate GWG and report the effect of insufficient and excessive GWG on pregnancy outcomes. 

 Adequate GWG now the reference group

 I would also suggest them to report whether the finding in the table three are crude or adjusted. If they have adjusted for confounders, they need to report the list of variables for which the analysis were adjusted.

 These are unadjusted estimates – indicated

 Similarly, the findings in table 4 were a bit confusing. No reference group. It is unclear what "-" mean.

 Table 4 now removed

 6. It would have been good if the authors reported how many of the participants were underweight, normal weight, overweight and obese. 

 Now reported.

 Underweight (36);; Normal weight (622), overweight (337), obese (232) 

7. It would be good if the authors commented on the implication and appropriateness of using the IOM recommendations for GWG for Nigerian women. 

 Comment added

 “The IOM guidelines are not yet in place, nor is there any consensus on GWG in Nigeria's maternal care [24, 28]. For instance, based on the IOM guidelines, the American College of Obstetricians and Gynaecologists recommends that clinicians calculate a woman's pre-pregnancy BMI at the first antenatal care visit and educate her on the importance of appropriate GWG goals [25]. Adopting and implementing a GWG guideline in Nigeria's maternal health services is essential. Although the IOM guideline is based on cut-off points from white women and is yet to be validated among Africans, the IOM guidelines are still the most utilised GWG guideline. Hence it could be helpful in Nigeria up until local guidelines are developed”

 Gestational weight gain was measured as the difference between the last third trimester weight recorded and the booking weight divided by the number of weeks between the weights. The authors need to substantiate the appropriateness of this GWG calculation method with evidence.

 The gestational age for booking weight was 20 weeks. Does this time have any implication of GWG?

---

## [Decision Letter · Decision Letter 1]

20 Jun 2023

PONE-D-22-35461R1Gestational Weight Gain among pregnant women in Ibadan, Nigeria: pattern, predictors and pregnancy outcomes.PLOS ONE

Dear Dr. Ikeola A. Adeoye,

Thank you for submitting your manuscript to PLOS ONE. After careful consideration, we feel that it has merit but does not fully meet PLOS ONE’s publication criteria as it currently stands. Therefore, we invite you to submit a revised version of the manuscript that addresses the points raised during the review process.

We look forward to receiving your revised manuscript.

Kind regards,

Malshani Lakshika Pathirathna, PhD

Academic Editor

PLOS ONE

Reviewers' comments:

Reviewer's Responses to Questions

**Comments to the Author**

1. If the authors have adequately addressed your comments raised in a previous round of review and you feel that this manuscript is now acceptable for publication, you may indicate that here to bypass the “Comments to the Author” section, enter your conflict of interest statement in the “Confidential to Editor” section, and submit your "Accept" recommendation.

Reviewer #1: (No Response)

Reviewer #2: (No Response)

2. Is the manuscript technically sound, and do the data support the conclusions?

Reviewer #1: Yes

Reviewer #2: Partly

3. Has the statistical analysis been performed appropriately and rigorously? 

Reviewer #1: Yes

Reviewer #2: No

4. Have the authors made all data underlying the findings in their manuscript fully available?

Reviewer #1: No

Reviewer #2: Yes

5. Is the manuscript presented in an intelligible fashion and written in standard English?

Reviewer #1: Yes

Reviewer #2: No

6. Review Comments to the Author

Reviewer #1: Please find below my comments:

1. Line 154: …..”

2. Line 154-155: The statement is not clear or scientifically precise. Suggesting “Furthermore, the current body of evidence on gestational weight gain (GWG) in Nigeria is limited, with only a few emerging studies available [29-32].”

3. Please include the write-up or description of Figure 1 in the methodology section, as it is important to mention that 1254 out of 1745 participants were included in the final analysis.

4. Line 199-200: Suggest rephrasing this statement to improve consistency in terms of presenting the time intervals and stages of follow-up for the study participants

5. Line 201-202: suggest including the mean and standard deviation for the week of gestation during the third trimester

6. Line 216: “gestational weight gain” Please ensure consistency in the usage of terms throughout the entire write-u

7. Line 218-220: It is recommended to rephrase the statement to align with scientific writing principles

8. Line 222-225: Suggest rephrasing these statements. It is important to mention that the Institute of Medicine (IOM) guidelines provide recommendations for weight gain during pregnancy based on a woman's pre-pregnancy body mass index (BMI). Consequently, it is relevant to connect this information to the subsequent statement

9. Line 248-249: please provide a reference for the birth weight categories. Same comments for gestational age at delivery, APGAR and Postpartum Haemorrhage

10. Line 272-274: Please include the “Forest plots showing the association of specific pregnancy outcomes with gestational weight gain (GWG) based on maternal BMI” in the statistical analysis section. Additionally, clarify the reason why the underweight group is not included in this analysis.

11. Line 288: Does the term "physical activity" refer to motorized transport in the table? If it does, please ensure consistent use of terms to avoid confusion.

12. Line 289-291: The statement, "Specifically, women with higher education and income experienced more excessive gestational weight gain (GWG) than insufficient or adequate GWG compared to women with lower education and income," is incorrect. It is important to note that chi-square tests only determine the overall association and do not specifically examine associations within specific categories. Therefore, this statement should be revised to avoid implying specific associations within subcategories based on chi-square analysis.

13. Line 295-296: The information mentioned cannot be found in the table.

14. Table 1:

a. “Prevalence” The term "prevalence" may not be suitable in this context. I suggest using the term "proportion" instead. "Prevalence" typically refers to the occurrence of a disease or illness, whereas adequate weight gain is not a disease but rather a desirable outcome. Therefore, using "proportion" would be more appropriate to describe the occurrence or proportion of participants with adequate weight gain.

b. Please check again the grouping for age (> 35 years), income or household income, motorized transport (yes/no or physically active/physically inactive)

c. Please rearrange the sequence of the variables, starting with socio-demographic variables and followed by lifestyle characteristics. Additionally, please ensure that the group categorizations are consistent with the write-up in lines 230-237. Ethnicity is not mentioned in the text.

d. Since new variables have been included in Table 1, it is suggested to add the corresponding write-up in the text

15. Figure 2: Please review the percentages again, as the total percentage for normal weight and overweight appears to exceed 100%

16. Line 352-355: It is suggested to also include the finding on excessive GWG.

17. Line 353-354: Suggest rephrasing to avoid starting the sentence with "and"

18. Line 375-383: Suggest adding a statement to mention that no significant associations were observed between pregnancy outcomes and gestational weight gain. This helps prevent potential misinterpretation or assumptions of associations that do not exist, ensuring the accuracy and reliability of the study's conclusions.

19. Line 386: “maternal body mass index”. Please use the abbreviation "BMI" for maternal body mass index, as it is important to maintain consistency throughout the text.

20. Figure 3:

a. Please ensure consistency by using the same term, such as “spontaneous vertex or spontaneous vaginal delivery” throughout the entire text. This will help avoid confusion and maintain clarity in the discussion.

21. Line 427-429: Please include reference for this statement

22. Line 435: 29.% formatting error

23. Line 450: "correct GWG," it would be better to use "interventions encompass creating awareness of the appropriate gestational weight gain (GWG) range."

24. Line 455: "Gestational weight gain" itself is not a problem, but it can be a risk factor for adverse outcomes if it falls into the category of inappropriate or excessive weight gain. Therefore, it would be more accurate to refer to the interventions as targeting the risk factors for inappropriate gestational weight gain rather than gestational weight gain itself. This clarification highlights the importance of addressing the factors that contribute to undesirable weight gain during pregnancy. risk factor for inappropriate GWG Please revise accordingly

25. Line 503: There is no discussion on SSB

26. Line 562-563: rephrase the statement as this study did not find any significant association between excessive GWG and outcomes

27. Line 504-511: The variable of physical activity in this study pertains specifically to motorized transport. However, it is essential to note that physical activity encompasses more than just transportation-related activities. Therefore, in the methodology section, it is necessary to provide a clear definition of physical activity that encompasses a broader range of activities. Additionally, it is important to describe how this motorized transport variable is linked to the subsequent statement. Providing references that support the chosen definition and the association between motorized transport and physical activity would further enhance the clarity and validity of the study.

28. Line 529-530: Please include the reference for this statement

29. Line 537-538: Please include a discussion on this

30. Line 539-540: Please revise the statement to remove the dash

31. Line 547: suggest rephrasing the statement as the inclusion of alcohol consumption as a psychological factor is incorrect

Reviewer #2: I would like to express my gratitude to the authors for revising the paper and resubmitting it. However, I have noticed that the authors only addressed certain parts of my comments. One specific example is my suggestion to substantiate their gestational weight gain calculation methods with evidence. I had recommended that they discuss the appropriateness of the chosen method in their specific context, its limitations, and support their claims by citing existing evidence. Although the authors attempted to address this in the response section, they failed to provide substantial evidence to support their claims. Furthermore, they neglected to discuss this matter in the manuscript itself.

Another concern I raised was regarding the implications of using booking weight at 20 weeks of gestation. Unfortunately, the authors did not respond to this concern or address the issue in the manuscript. It is important to note that 20 weeks of gestation falls in the middle of the second trimester, during which women typically experience a significant amount of weight gain that could potentially impact the calculation of gestational weight gain in the manuscript. This issue should have been clearly discussed, at the very least, as a limitation of the study.

Additionally, while the authors responded that the findings presented in Table 3 are crude relative risks, it is essential that this be clearly indicated in the manuscript, including within the table itself.

In Table 2, the authors calculated odds ratios to assess factors associated with insufficient or excessive gestational weight gain. However, they interpreted these findings as if they were relative risks. They used the term "likelihood," which is not appropriate when discussing odds ratios.

7. PLOS authors have the option to publish the peer review history of their article (what does this mean?). If published, this will include your full peer review and any attached files.

Reviewer #1: No

Reviewer #2: No

---

## [Author Response · Author response to Decision Letter 1]

10 Jul 2023

POINT BY POINT RESPONSE TO THE REVIEWERS COMMENTS

Reviewer 1

 Line 154: …..” 

 Done 

 Line 154-155: The statement is not clear or scientifically precise. Suggesting “Furthermore, the current body of evidence on gestational weight gain (GWG) in Nigeria is limited, with only a few emerging studies available [29-32].”

 Done

 Please include the write-up or description of Figure 1 in the methodology section, as it is important to mention that 1254 out of 1745 participants were included in the final analysis.

 description of Figure 1 included 

 The flowchart of study participants from enrolment to delivery is shown in fig 1 and 1254 out of the 1745 participants were included in the final analysis.

 Line 199-200: Suggest rephrasing this statement to improve consistency in terms of presenting the time intervals and stages of follow-up for the study participants

 Rephrased

 After recruitment (≤ 20 weeks), study participants were followed up at three time points: 24 -28 weeks, third trimester, at delivery.

 Line 201-202: suggest including the mean and standard deviation for the week of gestation during the third trimester

 the mean and standard deviation for the week of gestation during the third trimester included 

 Serial weight values were used to estimate GWG according to the Institute of Medicine guidelines (2009) [1] which was the difference between the booking weight and the last weight assessed during the third trimester at the ANC (32.4 ± 4.7 weeks) .

 Line 216: “gestational weight gain” Please ensure consistency in the usage of terms throughout the entire write-up

 Effected – GWG used throughout the manuscript after the first mention of “gestational weight gain”

 Line 218-220: It is recommended to rephrase the statement to align with scientific writing principles

 Rephrased

The GWG rate was calculated as the difference between the last third-trimester weight recorded and the booking weight divided by the number of gestational weeks between in the interval these weights were measured as annotated as follows. 

 GWG rate = (third trimester weight (final)- weight at booking)/(GA at third trimester weight (final)- GA at booking)

 Line 222-225: Suggest rephrasing these statements. It is important to mention that the Institute of Medicine (IOM) guidelines provide recommendations for weight gain during pregnancy based on a woman's pre-pregnancy body mass index (BMI). Consequently, it is relevant to connect this information to the subsequent statement

 DONE, Sir

 The Institute of Medicine (IOM) guidelines provide recommendations for weight gain during pregnancy based on a woman's pre-pregnancy body mass index (BMI). Hence, we categorised the GWG rate into insufficient, adequate, and excessive weight gain.

 Line 248-249: please provide a reference for the birth weight categories. Same comments for gestational age at delivery, APGAR and Postpartum Haemorrhage

 Provide references.

 Basu JK, Jeketera CM, Basu D. Obesity and its outcomes among pregnant South African women. International journal of gynaecology and obstetrics: the official organ of the International Federation of Gynaecology and Obstetrics. 2010;110(2):101-4.

 Sebire NJ, Jolly M, Harris JP, Wadsworth J, Joffe M, Beard RW, Regan L, Robinson S. Maternal obesity and pregnancy outcome: a study of 287,213 pregnancies in London. International journal of obesity and related metabolic disorders : journal of the International Association for the Study of Obesity. 2001;25(8):1175-82.

 Cnattingius S, Bergström R, Lipworth L, Kramer MS. Prepregnancy weight and the risk of adverse pregnancy outcomes. The New England journal of medicine. 1998;338(3):147-52.

 Ezeanochie MC, Ande AB, Olagbuji BN. Maternal obesity in early pregnancy and subsequent pregnancy outcome in a Nigerian population. African journal of reproductive health. 2011;15(4):55-9.

 Line 272-274: Please include the “Forest plots showing the association of specific pregnancy outcomes with gestational weight gain (GWG) based on maternal BMI” in the statistical analysis section. Additionally, clarify the reason why the underweight group is not included in this analysis.

 Added in the statistical analysis section as ….

 Forest plots showing the association of specific pregnancy outcomes with GWG based on maternal BMI” were also presented.

 why the underweight group

 Footnote for the forest plots

 Underweight weight was absent from the forest plots because they made up a very low proportion of the final analysis 36 (2.9%) and was mostly omitted from the output of the stratified analysis for the forest plots.

 Line 288: Does the term "physical activity" refer to motorized transport in the table? If it does, please ensure consistent use of terms to avoid confusion.

 Harmonized and defined

 Line 289-291: The statement, "Specifically, women with higher education and income experienced more excessive gestational weight gain (GWG) than insufficient or adequate GWG compared to women with lower education and income," is incorrect. It is important to note that chi-square tests only determine the overall association and do not specifically examine associations within specific categories. Therefore, this statement should be revised to avoid implying specific associations within subcategories based on chi-square analysis.

 Corrected and recast

 Specifically, the level of education and income had a positive association with excessive GWG i.e. women with primary education or less: 15 (45.5 %), secondary 151 (44.0 %), and tertiary 537 (61.4%) and income: < 20,000 - 192 (46.7%), "20,000 – 99,999", 373 (63.2%), and ≥ 100,000 48 (61.5%).

 Line 295-296: The information mentioned cannot be found in the table.

 Highlighted in table 1

 Also, more Christians (60.7%) than Muslims (49.6 %) experienced excessive GWG. Women excessive GWG spent significantly less time on moderate intensity activity per week (25.3±23.1) minutes, more physically inactive (60.7%) and more sedentary ( 61.0) than women with inadequate GWG who spent (28.4 ± 23.6) minutes/week on moderate intensity activity, less more physically inactive (52.0%) and less sedentary (52.3%). Conversely, women who experienced antepartum depression (APD) had a higher proportion of insufficient GWG (39.2%) compared to women without APD (24.9%)

 Table 1: 

 . “Prevalence” The term "prevalence" may not be suitable in this context. I suggest using the term "proportion" instead. "Prevalence" typically refers to the occurrence of a disease or illness, whereas adequate weight gain is not a disease but rather a desirable outcome. Therefore, using "proportion" would be more appropriate to describe the occurrence or proportion of participants with adequate weight gain.

 Changed to PROPORTION

 Please check again the grouping for age (> 35 years), income or household income, motorized transport (yes/no or physically active/physically inactive) 

 Grouping checked

 < 35 years (not > 35 years)

 Further details physical activity provided

 Please rearrange the sequence of the variables, starting with socio-demographic variables and followed by lifestyle characteristics. Additionally, please ensure that the group categorizations are consistent with the write-up in lines 230-237. Ethnicity is not mentioned in the text. 

 Since new variables have been included in Table 1, it is suggested to add the corresponding write-up in the text

 Ethnicity is now mentioned in the text. 

 The mean age of study participants was x ± SD and majority belonged to the Yoruba ethnic group (88.8%).

 Figure 2: Please review the percentages again, as the total percentage for normal weight and overweight appears to exceed 100%

 Errors now corrected all percentages now add up to 100%

 Line 352-355: It is suggested to also include the finding on excessive GWG.

 Findings included

 Specifically, the odds of excessive GWG was associated with increasing revenue' 20,000-99,999' (AOR: 1.64, 95% CI: 1.25 – 2.17), '≥ 100,000' (AOR: 1.54, 95 CI: 0.90 – 2.64), compared to those who earned less than 20,000. Compared with women with normal weight, women with overweight (AOR: 2.07, 95% CI 1.52 – 2.88) and obese (AOR: 1.56, 95% CI 1.11– 2.20) had a higher odds of excessive weight gain. Also women who had depression during pregnancy were also less likely to experience excessive GWG (AOR: 0.56, 95% CI 0.36 – 0.82). Conversely, compared with Christian women, Muslims had higher odds for insufficient GWG (AOR: 1.52, 95% CI 1.13 – 2.05). And women with APD had 70% higher odds of insufficient GWG (AOR: 1.70, 95% CI 1.17 – 2.45) than women without APD. 

 Line 353-354: Suggest rephrasing to avoid starting the sentence with "and"

 “And” is removed

 Line 375-383: Suggest adding a statement to mention that no significant associations were observed between pregnancy outcomes and gestational weight gain. This helps prevent potential misinterpretation or assumptions of associations that do not exist, ensuring the accuracy and reliability of the study's conclusions. 

 Added, thanks Sir

 There were no statistically significant associations observed between pregnancy outcomes and gestational weight gain.

 Line 386: “maternal body mass index”. Please use the abbreviation "BMI" for maternal body mass index, as it is important to maintain consistency throughout the text.

 "BMI" for maternal body mass index in the text

 Figure 3:

Please ensure consistency by using the same term, such as “spontaneous vertex or spontaneous vaginal delivery” throughout the entire text. This will help avoid confusion and maintain clarity in the discussion.

 “spontaneous vaginal delivery used throughout the manuscript”

 Line 427-429: Please include reference for this statement

 Canada, Australia and Brazil - DONE

 Line 435: 29.% formatting error

 DONE – 29.4%

 Line 450: "correct GWG," it would be better to use "interventions encompass creating awareness of the appropriate gestational weight gain (GWG) range."

 Change effected

 These interventions encompass creating awareness on the appropriate gestational weight gain (GWG) range,

 Line 455: "Gestational weight gain" itself is not a problem, but it can be a risk factor for adverse outcomes if it falls into the category of inappropriate or excessive weight gain. Therefore, it would be more accurate to refer to the interventions as targeting the risk factors for inappropriate gestational weight gain rather than gestational weight gain itself. This clarification highlights the importance of addressing the factors that contribute to undesirable weight gain during pregnancy. risk factor for inappropriate GWG Please revise accordingly

 Revised accordingly

 Investigating the risk factors associated with inappropriate GWG is essential for providing targeted interventions for addressing the factors that contribute to undesirable weight gain during pregnancy among Nigerian women.

 Line 503: There is no discussion on SSB

 Added

 In addition, although our study did not find any significant relationship between SSB intake and GWG, SSBs have been associated with obesity, weight gain and excessive GWG in the literature [2-4] because of the high and easily absorbed sugar content that leads to high energy intake and weight gain [5].

 Line 562-563: rephrase the statement as this study did not find any significant association between excessive GWG and outcomes

 DONE

 Even though the incidences of caesarean section and postpartum haemorrhage were higher among women with excessive GWG compared with adequate and insufficient GWG, we did not find any significant association between excessive GWG and pregnancy outcomes. 

 Line 504-511: The variable of physical activity in this study pertains specifically to motorized transport. However, it is essential to note that physical activity encompasses more than just transportation-related activities. Therefore, in the methodology section, it is necessary to provide a clear definition of physical activity that encompasses a broader range of activities. Additionally, it is important to describe how this motorized transport variable is linked to the subsequent statement. Providing references that support the chosen definition and the association between motorized transport and physical activity would further enhance the clarity and validity of the study. 

 ADDRESSED

 METHOD

 The participants’ level of physical activity is assesses using the (i.) passive transport (ownership of motorized transport) i.e. physically inactive while active transport or walking as physically active)[6, 7] (ii.) Duration of moderate-intensity exercise (minutes per week) is the total time spent on moderate-intensity activity per week[8]. (iii.) Sedentary behaviour was also assessed using the pregnancy physical activity questionnaire (PPAQ) and classified as high or low according to the PPAQ instruction guide [9, 10].

Physical Activity 

Physically active 635(50.6)

Physically inactive 619(49.4)

Sedentary Behaviour 

Low 561 (52.9)

High 518 (48.1)

Moderate intensity activity duration (minutes) 25.8±22.7

 RESULT

 Women excessive GWG spent significantly less time on moderate intensity activity per week (25.3±23.1) minutes, more physically inactive (60.7%) and more sedentary ( 61.0%) than women with inadequate GWG who spent (28.4 ± 23.6) minutes/week on moderate intensity activity, less more physically inactive (52.0%) and less sedentary (52.3%). 

 DISCUSSION

 We assessed physical activity using active/passive transport [6], duration of moderate-intensity exercise [6, 8] and sedentary behaviour [9, 10]. Active transport is a measure of physical activity because it contributes to total physical activity and increases energy expenditure [6, 7]. Women excessive GWG spent less time on moderate intensity activity per week minutes, more physically inactive and more sedentary than women with inadequate GWG. We found that physically in active and sedentary women had higher odds for excessive GWG than physically active women. Although, the relationship became insignificant on multivariate analysis. Importantly, the WHO recommends that pregnant women engage in at least 150 minutes of moderate-intensity physical activity during the week [8]. Researchers have reported poor compliance with the recommendation among Nigerian pregnant women [11, 12]. Hence the need to actively promote physical activity among pregnant women to obtain benefits, including improving cardiovascular fitness, preventing excessive GWG and GDM, improving sleep quality and so on [13-15].

 Line 529-530: Please include the reference for this statement

 DONE

 Line 537-538: Please include a discussion on this

 Discussion on Obesity and Post-partum haemorrhage is included

 The association between maternal obesity postpartum haemorrhage has been reported in the literature [16-18]. The plausible reasons include poor uterine contractility in obese women compared with non-obese [19], increase foetal weight that could lead to uterine atony or perineal tear, associate large placenta and placenta praevia [18]. Blomberg et al. in the cohort study among Swedish pregnancy reported that the risk of atonic uterine haemorrhage increased rapidly with BMI [20]. Therefore, extra vigilance is advised in the active management of the third stage of labour in obese women with excessive GWG [18, 19]. 

 Line 539-540: Please revise the statement to remove the dash

 DONE

 This study contributes significantly to a critical gap in maternal health literature in Nigeria, in which GWG has been scantly examined.

 Line 547: suggest rephrasing the statement as the inclusion of alcohol consumption as a psychological factor is incorrect

 DONE

 We investigated the influence of a broader range of variables; sociodemographic, lifestyle (alcohol consumption, tobacco exposure, SSB intake, physical activity and sleep duration), healthcare utilisation and psychological factors such as APD, which were lacking in previous studies.

Reviewer #2: 

I would like to express my gratitude to the authors for revising the paper and resubmitting it. However, I have noticed that the authors only addressed certain parts of my comments.

Thank you for your support for our publication we apologize for the omission which was a complete oversight 

1. One specific example is my suggestion to substantiate their gestational weight gain calculation methods with evidence. I had recommended that they discuss the appropriateness of the chosen method in their specific context, its limitations, and support their claims by citing existing evidence. Although the authors attempted to address this in the response section, they failed to provide substantial evidence to support their claims. Furthermore, they neglected to discuss this matter in the manuscript itself.

Issues are addressed as follows

o In this study, we assessed GWG using the rate of weight gain per week rather than the total weight gain, even though a more precise measure, because information required for the estimation total weight gain i.e. the weight just before the onset of labour and her pre-pregnancy weight, were not available in our study setting. Hence, in such a situation IOM guidelines recommends rate of weight gain per week assessed within the two and third trimester [1, 21, 22] which has also been put to use by other researchers in other settings [1, 21, 23, 24]. 

Rasmussen KM, Yaktine ALE. Weight Gain During Pregnancy: Reexamining the Guidelines.Institute of Medicine. In: Rasmussen KM, Yaktine AL, editors. Weight Gain During Pregnancy: Reexamining the Guidelines. Washington (DC): National Academies Press (US) Copyright © 2009, National Academy of Sciences.; 2009.

Siega-Riz AM, Bodnar LM, Stotland NE, Stang. The Current Understanding of Gestational Weight Gain among Women with Obesity and the Need for Future Research. NAM Perspectives Discussion Paper,. National Academy of Medicine, Washington, DC. 201

Drehmer M, Duncan BB, Kac G, Schmidt MI. Association of second and third trimester weight gain in pregnancy with maternal and fetal outcomes. PloS one. 2013;8(1):e54704.

Campos CAS, Malta MB, Neves PAR, Lourenço BH, Castro MC, Cardoso MA. Gestational weight gain, nutritional status and blood pressure in pregnant women. Revista de saude publica. 2019;53:57.

2. Another concern I raised was regarding the implications of using booking weight at 20 weeks of gestation. Unfortunately, the authors did not respond to this concern or address the issue in the manuscript. It is important to note that 20 weeks of gestation falls in the middle of the second trimester, during which women typically experience a significant amount of weight gain that could potentially impact the calculation of gestational weight gain in the manuscript. This issue should have been clearly discussed, at the very least, as a limitation of the study.

Issues addressed in the limitation

o In this study, we could not estimate total GWG because prepregnancy and delivery weights were unavailable; hence, we used a weekly GWG rate. Also, the use of booking weight at ≤ 20 weeks (mean 16.2 ± weeks) women would have experienced some weight gain could have potentially influenced gestational weight gain estimate. Therefore, GWG rate used in this study was a more appropriate measure as recommended by IOM guidelines [1]. 

3. Additionally, while the authors responded that the findings presented in Table 3 are crude relative risks, it is essential that this be clearly indicated in the manuscript, including within the table itself.

Unadjusted indicated in Table 3 and its description



4. In Table 2, the authors calculated odds ratios to assess factors associated with insufficient or excessive gestational weight gain. However, they interpreted these findings as if they were relative risks. They used the term "likelihood," which is not appropriate when discussing odds ratios.

Likelihood is replaced with odds throughout the manuscript

---

## [Decision Letter · Decision Letter 2]

2 Aug 2023

Gestational Weight Gain among pregnant women in Ibadan, Nigeria: pattern, predictors and pregnancy outcomes.

PONE-D-22-35461R2

Dear Dr. Ikeola A. Adeoye, 

We’re pleased to inform you that your manuscript has been judged scientifically suitable for publication and will be formally accepted for publication once it meets all outstanding technical requirements.

Kind regards,

Malshani Lakshika Pathirathna, PhD

Academic Editor

PLOS ONE

Additional Editor Comments (optional):

Reviewers' comments:

Reviewer's Responses to Questions

**Comments to the Author**

1. If the authors have adequately addressed your comments raised in a previous round of review and you feel that this manuscript is now acceptable for publication, you may indicate that here to bypass the “Comments to the Author” section, enter your conflict of interest statement in the “Confidential to Editor” section, and submit your "Accept" recommendation.

Reviewer #1: (No Response)

Reviewer #2: All comments have been addressed

2. Is the manuscript technically sound, and do the data support the conclusions?

Reviewer #1: Yes

Reviewer #2: Yes

3. Has the statistical analysis been performed appropriately and rigorously? 

Reviewer #1: Yes

Reviewer #2: Yes

4. Have the authors made all data underlying the findings in their manuscript fully available?

Reviewer #1: No

Reviewer #2: Yes

5. Is the manuscript presented in an intelligible fashion and written in standard English?

Reviewer #1: Yes

Reviewer #2: Yes

6. Review Comments to the Author

Reviewer #1: Dear author, thank you for the amendments. However, I still have some concerns regarding the manuscript. Please find below my comments:

1. Line 222-233: I would like to kindly request the authors to recheck the formula for the GWG rate. According to the IOM (Institute of Medicine) recommendation, the GWG rate is calculated as the difference between the weight at delivery and the weight at the beginning of the 2nd trimester, divided by the number of weeks of gestation. If the authors prefer to use the existing formula, please provide a reference for it, as well as the categorization for the cut-off?

Reviewer #2: I would like to say thank you for addressing my comments. I don not have additional comments.

7. PLOS authors have the option to publish the peer review history of their article (what does this mean?). If published, this will include your full peer review and any attached files.

Reviewer #1: No

Reviewer #2: No

---

## [Editor Report · Acceptance letter]

8 Aug 2023

PONE-D-22-35461R2 

Gestational weight Gain among pregnant women in Ibadan, Nigeria: pattern, predictors and pregnancy outcomes. 

Dear Dr. Adeoye:

I'm pleased to inform you that your manuscript has been deemed suitable for publication in PLOS ONE. Congratulations! Your manuscript is now with our production department. 

Kind regards, 

on behalf of

Dr. Malshani Lakshika Pathirathna 

Academic Editor

PLOS ONE